# The highly dynamic pangenome of basal chordates is enriched in defence and immunity genes and is inherited following the Mendelian law

Umberto Rosani[1]*, Marco Gerdol[2], Mart Krupovic[3]

1 Department of Biology, University of Padova, Padova, Italy, 2 Department of Life Sciences, University of Trieste, Trieste, Italy, 3 Institut Pasteur, Université Paris Cité, CNRS UMR6047, Cell Biology and Virology of Archaea Unit, Paris, France

* umberto.rosani@unipd.it

## Abstract

Pangenome analyses, which encompass the full genetic repertoire of a species, offer valuable insights into intraspecific diversity and phylogeographic gene patterns. While the taxonomic breadth and functional significance of animal pangenomes remain to be fully uncovered, recent findings (such as reports of open, bacterial-like pangenomes in bivalves) highlight the need to better understand the molecular mechanisms driving inter-haplotype structural variation. Genes affected by presence-absence variation (PAV), along with non-reference sequences (NRSs), represent evolutionary footprints that may shape genome architecture and plasticity, ultimately influencing the adaptability and long-term fitness of species. To investigate the pangenomic architecture of basal chordates, we analyzed available whole-genome resequencing data from *Branchiostoma belcheri* and *B. floridae*, examined the impact of structural genomic variation, and assessed the inheritance patterns of dispensable genes across generations. The pangenomes of both species include over a thousand of genes affected by PAV and exhibiting trans-generational Mendelian transmission from parents to offspring. We further demonstrate that 35 dispensable genes in *B. belcheri* are of exogenous origin, likely resulting from the integration of a malacoherpesvirus genome, thereby extending the known host range of *Malacoherpesviridae* from invertebrates to chordates. PAV preferentially targeted gene families involved in defense, immunity, and cell signaling, including GTPases of immunity-associated proteins (GIMAPs), caspases, toll-like receptors, and pattern recognition receptors containing apextrin C-terminal (APEC) domains. The dynamic nature of immunity genes in cephalochordates parallels patterns seen in open bacterial pangenomes, suggesting that fundamental principles of genome evolution and innovation across life domains are shaped by host–pathogen interactions.

**Data availability statement:** All the data are either already available (as indicated in the M&M section) or included in the Supporting information.

**Funding:** This work was supported by the Italian National Project PRIN PNRR 2022 (ID: Prot. P2022JEEMT - Developing a tool for the study of haplotype diversity in Mytilus galloprovincialis), which provides support to UR and MG. The funder had no role in study design, data collection and analysis, decision to publish, or preparation of the manuscript.

**Competing interests:** The authors have declared that no competing interests exist.

## Author summary

Understanding how genetic diversity arises and is maintained within a species is key to reveal evolution and adaptation mechanisms. This study explores the full set of genes - or "pangenome" - in two species of lancelets (*Branchiostoma belcheri* and *B. floridae*), an early-diverging lineage of chordates. By analysing whole-genome sequences, we found that more than a thousand genes are not shared by all individuals. These dispensable genes were passed from parent to offspring like typical inherited traits and are enriched in domains involved in immunity and cell signaling. Strikingly, a block of 35 dispensable genes in *B. belcheri* represented an integrated genome of a malacoherpesvirus, revealing both the integration event and supporting the association between these viruses and lancelets. Overall, our findings revealed that immune-related genes are especially prone to be part of the dispensable gene set in both lancelet species, similar to what is observed in bacteria. This observation highlights a shared evolutionary strategy across life domains, where genetic flexibility - especially in defensive genes - may be crucial for survival in a changing environment.

## Introduction

Biodiversity is a unifying concept in biology that can be studied on different levels, ranging from genes to ecosystems [1]. Indeed, although the definition of biodiversity is mostly discussed at the level of species, it is grounded in the genetic variation within species [2], and is tightly connected to the concept of function, which also spans the molecules-to-ecosystem range [3]. First proposed by Tetz in 2003, the term pangenome, according to its original definition, was intended as a unifying view of all genetic information, including the complete gene set of all living organisms, along with associated viruses and mobile genetic elements (MGE) [4,5]. However, in subsequent years, this term has been mostly used in the scientific literature with a much narrower scope, referring specifically to the full set of genes found across all strains of a given bacterial species. These genes can be classified as core, when shared by all individuals, or as dispensable, when subject to presence-absence variation (PAV) among individuals [6]. Dispensable genes, also referred to as 'auxiliary' genes [7], can be further classified as softcore, shell or cloud, based on their decreasing frequency in a population. The pangenome of each species can be defined as 'open' or 'closed', depending on the increase (or lack thereof) in the number of novel genes provided by sequencing of additional individuals [8]. Although this gene-centric definition derives from the fact that early studies on pangenomes were primarily conducted in prokaryotes [9], the subsequent identification of pangenomic architectures in eukaryotes prompted a broader reinterpretation. Today, the term pangenome is not limited to protein-coding genes but usually encompasses a wide range of genetic elements, more accurately capturing the full extent of intraspecific genomic diversity. Nevertheless, gene-centric approaches continue to be a key element in pangenomic

studies, because they allow obtaining a direct link between dispensable genes, their associated functions and, eventually, phenotypic traits [10]. Besides bacteria, pangenomes have been reported in plants [11], archaea [12], giant viruses [13], fungi [14] and animals, including humans [15], goat [16] and, among invertebrates, in the bivalve *Mytilus galloprovincialis* [17]. Although open pangenomes are more common in bacteria than eukaryotes [8,18,19], the relationships among pangenome type, genome size, repeat content, and species phylogeny remain poorly understood [10]. In bacterial pangenomes, the interplay between horizontal gene transfer (HGT) and homologous recombination likely shapes genome composition, with inter-species boundaries and ecological barriers to DNA exchange acting as key determinants [20]. Notably, the dispensable fraction of bacterial pangenomes is strongly enriched in genes involved in defence mechanisms and mobile genetic elements [21–23].

In sexual eukaryotes, where stronger evolutionary constraints typically limit genomic plasticity, pangenome evolution may be primarily shaped by sexual recombination, gene duplication, and domain reshuffling, with lateral gene transfers playing a comparatively minor role [24,25]. However, over long evolutionary timescales, mechanisms such as virus-mediated gene transfer, retrotransposon activity, homology-driven recombination, and non-homologous end joining may significantly contribute to the emergence and maintenance of dispensable genomic content [25,26]. Although the genomic mechanisms responsible for the emergence of dispensable genes are increasingly recognized, much less attention has been given to the transmission dynamics of these genes across generations. Dispensable genes can be modelled as biallelic loci, where one allele corresponds to gene presence and the other to gene absence, and are therefore assumed to follow Mendelian inheritance, segregating normally among offspring. However, this assumption may not always hold, because dispensable genes are frequently embedded within large structural variants that may impair homologous chromosome pairing during meiosis [27]. From a theoretical standpoint, such disruptions can lead to segregation distortion or non-random transmission, resulting in deviations from expected Mendelian ratios in progeny. Additionally, while each dispensable gene can be individually absent, the combined absence of multiple functionally redundant dispensable genes may not be equally tolerated. In such cases, selective constraints may limit the viable gene PAV configurations across individuals. In this context, although pangenomic studies in metazoans are still in their infancy, recent reports strongly suggest that transposable elements play a key role in driving the acquisition and maintenance of intraspecific structural variation [28]. Integration and endogenization of viral genomes within the host chromosomes further expands the functional landscape and complexity of pangenomes. This observation is supported by the identification of non-reference sequences (NRSs), likely originating from the combined activity of diverse MGEs, including both transposons and bona fide capsid-forming viruses, in human resequencing datasets [29].

Lancelets are marine benthic animals that display cosmopolitan distribution in temperate waters and occupy a basal taxonomic position within the phylum Chordata. They have been used as a model to study the evolution of early vertebrates [30], as these two lineages share a common ancestor that lived around half a billion years ago [31], but the intraspecific structural genomic variations in lancelets have never been investigated. The recent expansion of the genomic datasets for cephalochordates provides a solid basis for pangenome investigations. Lancelets have received considerable attention due to the so-called 'amphioxus polymorphism paradox': despite the remarkably high nucleotide variability (1 SNP every 34 nt), no apparent morphological differences are present among individuals, suggesting that in this species, SNPs do not impact phenotypic characteristics [32]. Chromosome-level genome assemblies have been generated for *B. belcheri*, *B. japonicum* and *B. floridae* [33], as well as for *B. lanceolatum* [34], and high-coverage resequencing data has been generated for *B. belcheri*.

Here, we investigate the pangenomic architectures in two lancelet species, *B. floridae* and *B. belcheri*. We report the widespread occurrence of dispensable genomic regions affected by PAV and show that these regions are enriched in genes involved in immunity and signalling. These observations highlight parallels between the pangenomes of prokaryotes and lancelets, suggesting that genome dynamics in these organisms are shaped, to a considerable extent, by antagonistic host–pathogen interactions. Finally, we traced the NRSs in lancelet pangenomes and discovered the integration of

a nearly complete malacoherpesvirus genome into the *B. belcheri* chromosome, possibly representing an active lancelet virus with ancient connections to mollusk herpesviruses.

## Results

### Lancelet genomes are characterized by widespread PAV

As we have previously demonstrated [17,35], aligning whole-genome resequencing data to an annotated reference genome represents a robust approach for detecting PAV. Unlike core genomic regions, which are consistently shared among all individuals of a species, regions affected by PAV exhibit inter-individual structural variation and are often entirely absent or present in a hemizygous state. These regions, along with the genes they contain, are therefore considered dispensable and constitute the variable fraction of the pangenome of a species. We applied this approach to 99 *B. belcheri* and 41 *B. floridae* whole genome resequencing samples (S1 Table), revealing that a considerable fraction of the two reference genome assemblies, i.e., 16.1% and 10.3% respectively, were dispensable. Most of these regions were softcore (i.e., dispensable, but present in > 90% of the samples), whereas shell/cloud regions, absent in more than 10% of the considered individuals, only accounted for 3.7% and 6.6% of the genome size in *B. belcheri* and *B. floridae*, respectively (Table 1).

The analysis of protein-coding genes confirmed these results, identifying 3,171 and 2,309 dispensable genes in *B. belcheri* and *B. floridae*, respectively. As in the case of genomic regions, a significant fraction of dispensable genes could be assigned to the softcore category, and only 1,112 (in *B. belcheri*) and 1,281 (in *B. floridae*) genes were classified into the shell/cloud category (accounting for 4.4% of the total in both species) (Table 1, S1 and S2 Files and S1 Fig). The number of missing genes per genome ranged from 350 to 550 in *B. belcheri* and from 450 to 600 in *B. floridae* (Fig 1a). By contrast, the two gonad samples used to generate the reference genome assembly of *B. belcheri* possesses a few missing genes and regions and provided a benchmark for the false positive rate of our pipeline, which was estimated to be ~ 0.23% for genomic regions and 0.05-0.09% for genes. In both species, the number of missing genomic regions and genes per resequenced genome were strongly correlated (R = 0.58 in *B. belcheri*, R = 0.55 in *B. floridae*, Fig 1b).

In line with the expected lower frequency of occurrence of dispensable genes in lancelet populations, the analysis of the transcriptomic datasets available for both species (S1 Table) showed that the shell and cloud genes were expressed at significantly lower levels than the core genes (Fig 1c-d), to the point that their transcription was frequently undetectable (Fig 1e-f).

### Structural organization of lancet pangenomes

Despite the high quality of the available genomic resources for *B. floridae* and *B. belcheri*, the two reference genome assemblies are not haplotype-resolved, which limits the ability to directly assess the spatial and size distribution of structural variants through the alignment between haplotypes. Nonetheless, we leveraged the information derived from the analysis of PAV patterns presented in the previous section to investigate some of these aspects. First, the mapping of shell and cloud genes in *B. floridae*, the only of the two considered species with a chromosome-scale genome assembly, allowed us to visualize the chromosomal distribution of dispensable genes (Fig 2). These genes were found to be distributed across all 19 chromosomes and, although some regions showed variation in density, this pattern was consistent with a random distribution of structural variants, as no chromosome exhibited a significantly higher concentration of PAV-affected genes than the genome-wide average.

**Table 1. Results of the PAV analysis in the two lancelet species.**

| Species | No. of samples | Dispensable regions (No./Percentage) | Shell/cloud regions (No./Percentage) | Dispensable genes (No./Percentage) | Shell/cloud genes (No./Percentage) |
|---|---|---|---|---|---|
| *B. belcheri* | 99 | 13,932 (16.1%) | 3,228 (3.7%) | 3,171 (12.6%) | 1,112 (4.4%) |
| *B. floridae* | 41 | 10,625 (10.3%) | 6,805 (6.6%) | 2,309 (7.9%) | 1,281 (4.4%) |

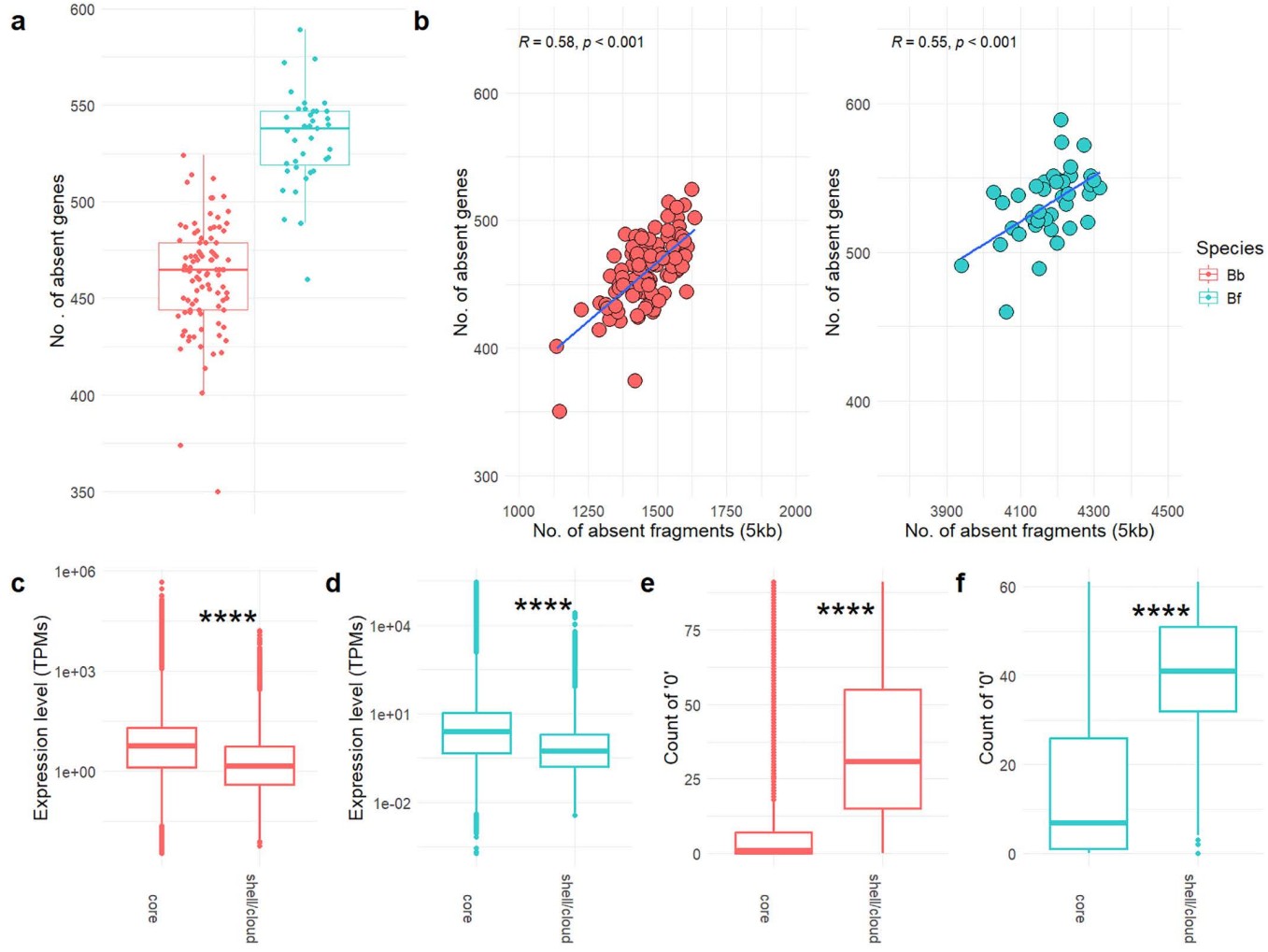

**Fig 1. PAV analysis of *B. belcheri* (red) and *B. floridae* (blue) whole genome resequencing datasets.** **a.** Number of absent genes per sample. **b.** Correlation between the number of absent genomic regions vs. the number of absent genes per sample, divided by species. The two DNA samples used to build the reference genome have been removed from the plot. c and **d.** Comparison between the expression levels of shell/cloud and core genes observed in the available RNA-seq datasets of *B. belcheri* (c) and *B. floridae* (**d**). e and **f.** Fraction of shell/cloud and core genes showing null expression values in the available RNA-seq datasets of *B. belcheri* (e) and *B. floridae* (**f**). ****, p-value <0.001.

We also analysed the distribution of contiguous shell/core gene block sizes in both species, finding that most of these consisted of singletons (80.4% in *B. belcheri*, 74.9% in *B. floridae*) or contained fewer than 5 genes (99.0% in *B. belcheri*, 99.1% in *B. floridae*) (S2 Table). This result aligns with the findings obtained from the correlation analysis of concordant PAV patterns and pairwise distances between pairs of dispensable genes placed on the same chromosome or scaffold. In fact, this correlation was significantly higher than random association only for distances below approximately 30 kb in *B. belcheri* and 25 kb in *B. floridae*, even though significant correlations were occasionally also observed across greater distances (S2 and S3 Figs). Thus, in both species of amphioxus, most of the genomic regions subjected to PAV were relatively small in size, with only rare instances of multiple neighbouring dispensable genes.

Nevertheless, an exception to this general pattern was observed in *B. belcheri*, which contains a block of 35 contiguous dispensable genes located on the same scaffold, spanning ~150 kb of sequence. Interestingly, these genes showed

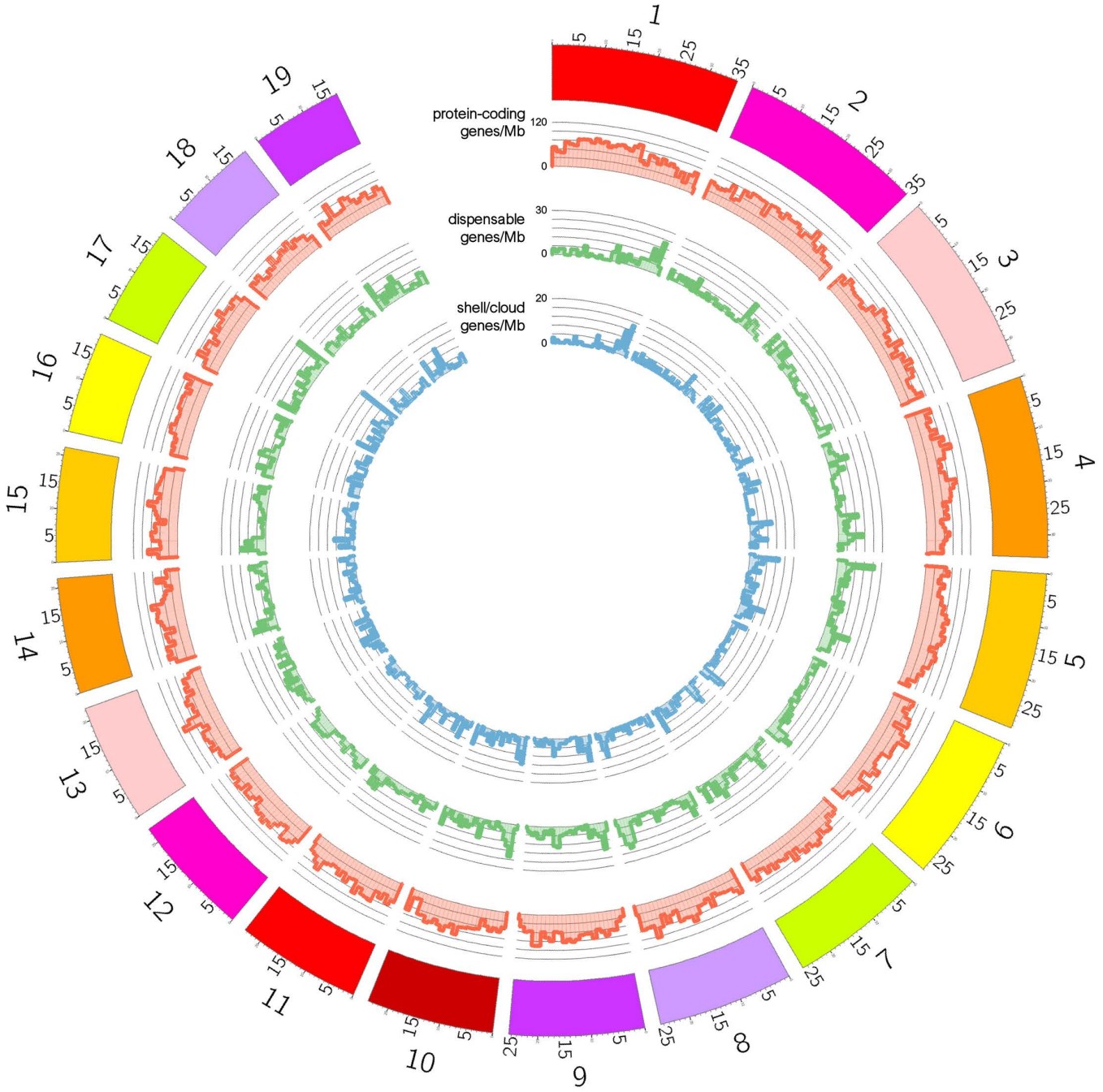

**Fig 2. Spatial distribution of shell/cloud genes, dispensable genes and protein coding genes in the genome of *B. floridae*, reported as gene density per Mb.** Note that unplaced genomic scaffolds and the mitochondrial genome were ignored.

highly concordant PAV patterns, being simultaneously absent in 79 out of the 99 resequenced individuals (S1 Fig), which resulted in a clear clustering of the resequenced genomes into two groups (S4 Fig). The nature of this unusually large dispensable region identified in the *B. belcheri* genome will be examined in detail in the following section.

Considering recent data supporting the key role played by transposable elements in establishing structural variation in the pangenome of bivalve mollusks [28], we investigated the potential enrichment of specific classes of repetitive elements in the regions flanking the shell and cloud gene blocks, as compared to the core genes. Although this analysis did not reveal a robust association between dispensable genomic regions and flanking MGEs, a significant enrichment of Gipsy-like LTR retrotransposons/reverse-transcribing viruses (family *Metaviridae*) and PIF/Harbinger-family DNA transposons was detected in *B. floridae* and *B. belcheri*, respectively, even though they were only associated with a very small fraction of all shell/cloud genes (i.e., ~ 1%) (S3 Table).

### An integrated malacoherpesvirus-like genome and lancelet NRS

As mentioned in the previous section, the in-depth analysis of gene PAV patterns led to the identification of an anomalous large block of dispensable genes in the *B. belcheri* reference genome, which was also detectable in 20 of the analysed resequenced samples. These genes were found to be part of the integrated genome of a herpes-like virus spanning 149,856 bp in length. The viral genome displayed a GC content of 45.7%, slightly higher than the average GC% of *B. belcheri* (41.3%), and was flanked by regions rich in repetitive elements, both at the 3'- (4.7 kb with 43% of repeats) and at the 5'-end (28.8 kb with 65.5% of repeats, Fig 3a and S3 File).

The presence of a putative bona fide virus was further supported by the reannotation of this region using criteria valid for viral genomes, yielding a total of 62 open reading frames (ORFs), 27 of which showed homology to malacoherpesviruses and four to vertebrate herpesviruses. Overall, 72.7% of the provirus region length was covered by protein-coding genes (Tables 2 and S4). Hereinafter, we refer to this integrated herpesvirus (HV) sequence in *B. belcheri* as BelcheriHV-1. The viral genome consists of three discontinuous gene blocks. The first block of 46 kb is characterized by tightly clustered viral genes (coding density: 82.8%) and was identified in 20/20 samples. The second block (21 kb) showed a sparser distribution of genes that were not recognizably similar to genes of known malacoherpesviruses (coding density: 37.1%). The latter gene block was covered by multiple assembled scaffolds in each resequenced sample, likely because of the presence of repeated regions. The last block, originally spanning 81.5 kb in length, showed a deletion of 5 kb in all the 20 samples, resulting in a length of 75.5 kb, with a coding density of 75.6% (Table 2). Within this block, malacoherpesvirus genes are intermixed with genes of unknown origin. Unfortunately, the junctions between blocks cannot be resolved based on the available data. Interestinlgy, the SNPs located on BelcheriHV-1 genome were mostly detected either with a frequency of 100% (fixed variations) or 50%, possibly indicating the presence of two BelcheriHV-1 variants in some of the tested genomes (Fig 3b). Overall, BelcheriHV-1 was 24 × less impacted by nucleotide variations than the *B. belcheri* genome, with an average of one SNP per 5,958 nt, vs one SNP per 174 nt, respectively (Fig 3c). This result strongly suggests that BelcheriHV-1 has evaded the 'amphioxus polymorphism paradox'.

The average sequencing coverage of BelcheriHV-1 in the 20 positive lancelet samples was slightly lower than the genome coverage (i.e., 0.7×), which is in line with the differences observed between the average coverage of the 35 predicted viral genes compared with the average coverage of the 23,868 lancelet genes (Fig 3d). Only a single muscle sample (SRR8324690) showed a coverage as high, i.e., 1.2 ×, whereas a gonad sample showed 0.9 × coverage based on genome regions (0.8× considering the gene coverages, Fig 3d).

We further screened the taxonomic composition of individual de novo assembled lancelet NRSs, which allowed the identification of three contigs showing similarity with BelcheriHV-1 obtained from the assembly of the genome of a single male *Belcheri* individual (SRR14996628), which tested negative for BelcheriHV-1. These contigs cumulatively amounted to 132.5 kb, encoded 72 ORFs (Tables 2 and S4 and S4 File) and showed homogenous coverage, which suggested that they were part of the same viral genome, called BelcheriHV-2. Both GC% content and coding density were similar to those of BelcheriHV-1 as well as to the gastropod-infecting malacoherpesvirus (HaHV-1, Table 2). The proviral genome was not present in any of the 41 available resequenced *B. floridae* samples or in any of the lancelet genome assemblies.

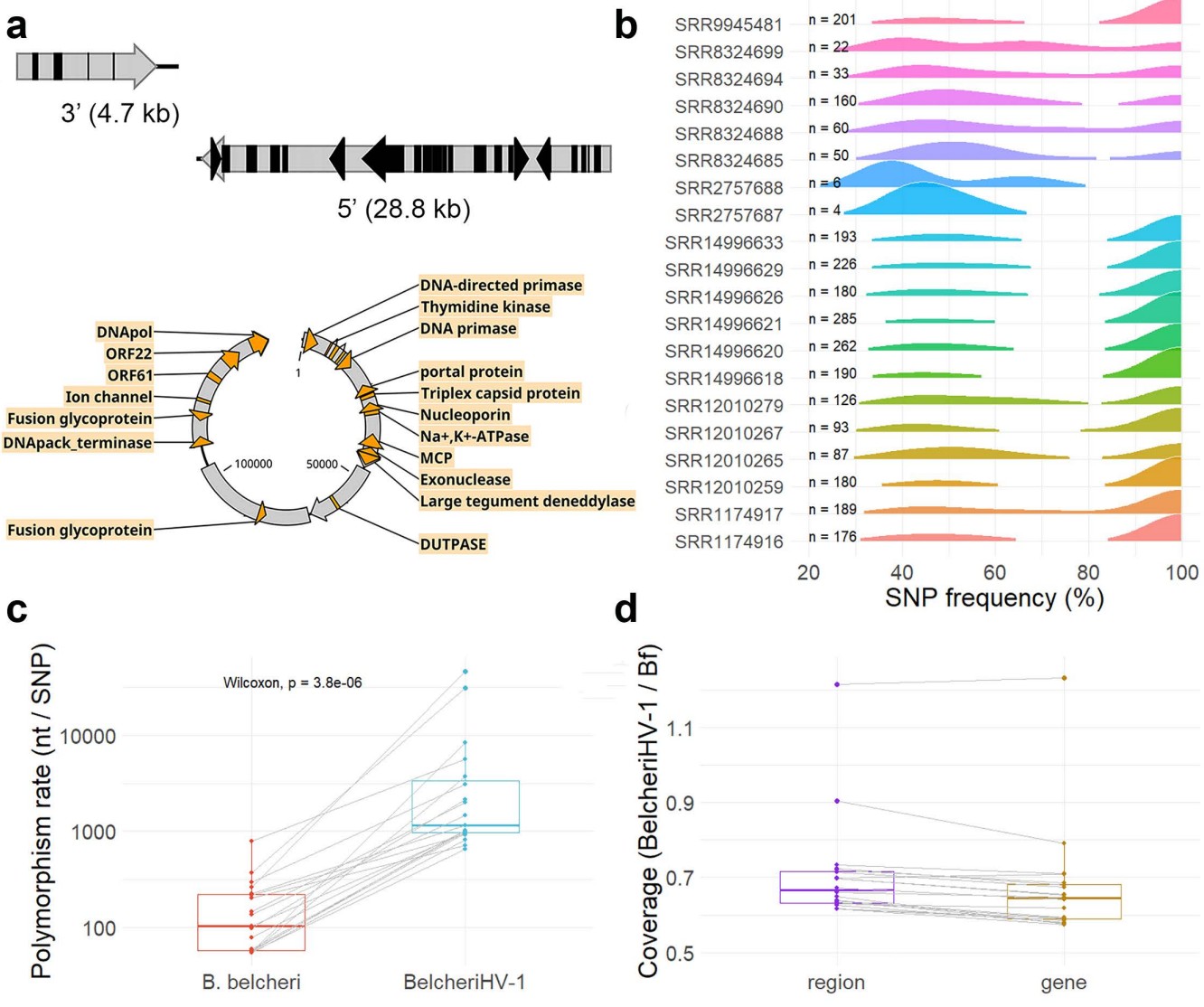

**Fig 3. The integration of BelcheriHV-1 into the genome of *B. belcheri*. a.** Circular visualization of the BelcheriHV-1 genome is provided, indicating its putative integration site between two genomic regions rich in interspersed repeats (reported as black arrows or boxes). In the BelcheriHV-1 genomes, the genes showing similarity with herpesviruses are reported as orange arrows, together with their putative annotations. More detailed annotations are reported in Tables 3 and S4. **b.** The distribution frequencies of the BelcheriHV-1 SNPs detected in the 20 positive samples. The total number of SNPs is also reported for each sample. **c.** The polymorphism rates, computed as the distance between SNPs, of *B. belcheri* and of BelcheriHV-1 in the 20 positive samples (the same samples are matched by grey lines). **d.** The ratio between the BelcheriHV-1 and the *B. belcheri* coverages computed based on non-overlapping sliding windows (region) or genes (the same samples are matched by grey lines).

## Dispensable genes are enriched in specific conserved domains and functions

To investigate the possible functional implications of lancelet dispensable genes, we analysed the predicted functions of the shell/cloud dispensable genes using an enrichment test based on the Pfam conserved domains and the Gene Ontology (GO) terms. A total of 46 and 26 conserved domains were significantly enriched among the *B. belcheri* and *B. floridae* shell/cloud dispensable genes, respectively (Fig 4a-b), with 17 domains shared by both species (Fig 4c and S5 Table).

**Table 2. Belcheri herpesviruses.** The table provided a comparison between the genome length, the number of predicted genes, the GC content and the coding density of belcheri herpesviruses in comparison with other two malacoherpesviruses (OsHV-1 and HaHV-1). For BelcheriHV1 the details regarding the three identified genome blocks are also reported.

| Virus name | Genome length | Host species | Status | No. of predicted genes | GC content (%) | Coding density (%) |
|---|---|---|---|---|---|---|
| BelcheriHV-1 | 149,856 | Lancelet | Integrated | 62 | 45.7 | 70.4 |
| -block 1 | 46,000 | / | / | 19 | 46.5 | 82.8 |
| -block 2 | 21,000 | / | / | 13 | 45.4 | 37.1 |
| -block 3 | 75,500 | / | / | 30 | 45.5 | 75.6 |
| BelcheriHV-2 | 132,488 | Lancelet | Integrated/fragmented | 74 | 44.9 | 85.1 |
| OsHV-1 | 203,893 | Bivalve | Free living | 127 | 38.6 | 83.8 |
| HaHV-1 | 210,993 | Gastropod | Free living | 114 | 46.9 | 81.6 |

In both species, the most over-represented was the Death domain (PF00531), present in 35 and 34 dispensable genes in *B. floridae* and *B. belcheri*, respectively. Several domains, typically implicated in the recognition of microbe-associated molecular patterns, such as the *lectin C-type domain* (PF00059), the *apextrin C-terminal domain* (PF16977) and the *TIR domain* (PF13676), as well as with intracellular immune signalling, such as *AIG1* (PF04548), were also significantly over-represented in both lancelet species.

The same analysis performed on the GO terms revealed 17 biological processes, 27 molecular functions and two cellular components (kinesin complex and microtubule) enriched in *B. belcheri*, whereas 16 biological processes, 13 molecular functions and one cellular component (plasma membrane) were enriched in *B. floridae* (Fig 4d-e and S5 Table). Notably, nine biological functions (signal transduction, regulation of apoptotic process, DNA integration, toll-like receptor signaling pathway, immune response, DNA recombination, positive regulation of interleukin-1 production, proteolysis and cell adhesion) and five molecular functions (GTP binding, cysteine-type endopeptidase activity, channel activity, cysteine-type peptidase activity, and protein tyrosine kinase activity) were shared between the two species (Fig 4f). Considering specific functional domains, the *lectin C-type* domain, responsible for carbohydrate binding and functioning as a receptor in various contexts, including adhesion and signalling, and the *death* domain, involved in apoptosis and inflammation, were the most enriched domains in *B. belcheri*. Similarly, *death* and *death effector* domains were the most enriched domains in *B. floridae*. The *MAC/perforin-like* domain found in cytolytic proteins that form pores in the target membranes, the *AIG1 family* domain present in the GTPase of the immunity associated protein (GIMAPs) family, a group of proteins involved in immunity in plants and invertebrates, and the *apextrin C-terminal domain* (APEC), associated with pattern recognition receptors, were all enriched in both lancelet species. The dispensability rates of the shell/cloud genes encoding selected Pfam domains were similar in the two species, except for the AIG1 family domain, which showed a higher dispensability in *B. belcheri* (Fig 4g). Since the size of most de novo assembled NRSs obtained from whole genome resequencing datasets was significantly smaller than the average gene length of both species (i.e., ~6 kb), most of the predicted genes associated with these sequences were expected to be fragmented. Despite this limitation, the analysis of the NRS-encoded proteins confirmed, in both species, the high representation of the same Pfam domains enriched among the shell/cloud dispensable genes (Fig 4h-i).

We further investigated the evolutionary history of the shell/cloud dispensable genes annotated in the reference genomes and those associated with NRSs by determining their orthology based on 11 selected proteomes including three lancelet, one colonial ascidia, three vertebrate and three invertebrate species (Fig 5). We retrieved 1,021 orthogroups that included dispensable genes of *B. belcheri* (N = 564) or *B. floridae* (N = 699). Dispensable genes were mostly present in different combinations among lancelet species, and only in 4.4% of the cases they belonged to orthogroups with a broad taxonomic distribution, covering all the other considered species (Fig 5a). Notably, 243 of orthogroups are shared between *B. belcheri* and *B. floridae* (Fig 5b). On average 49% and 33% of the paralogs per orthogroup were dispensable in *B.*

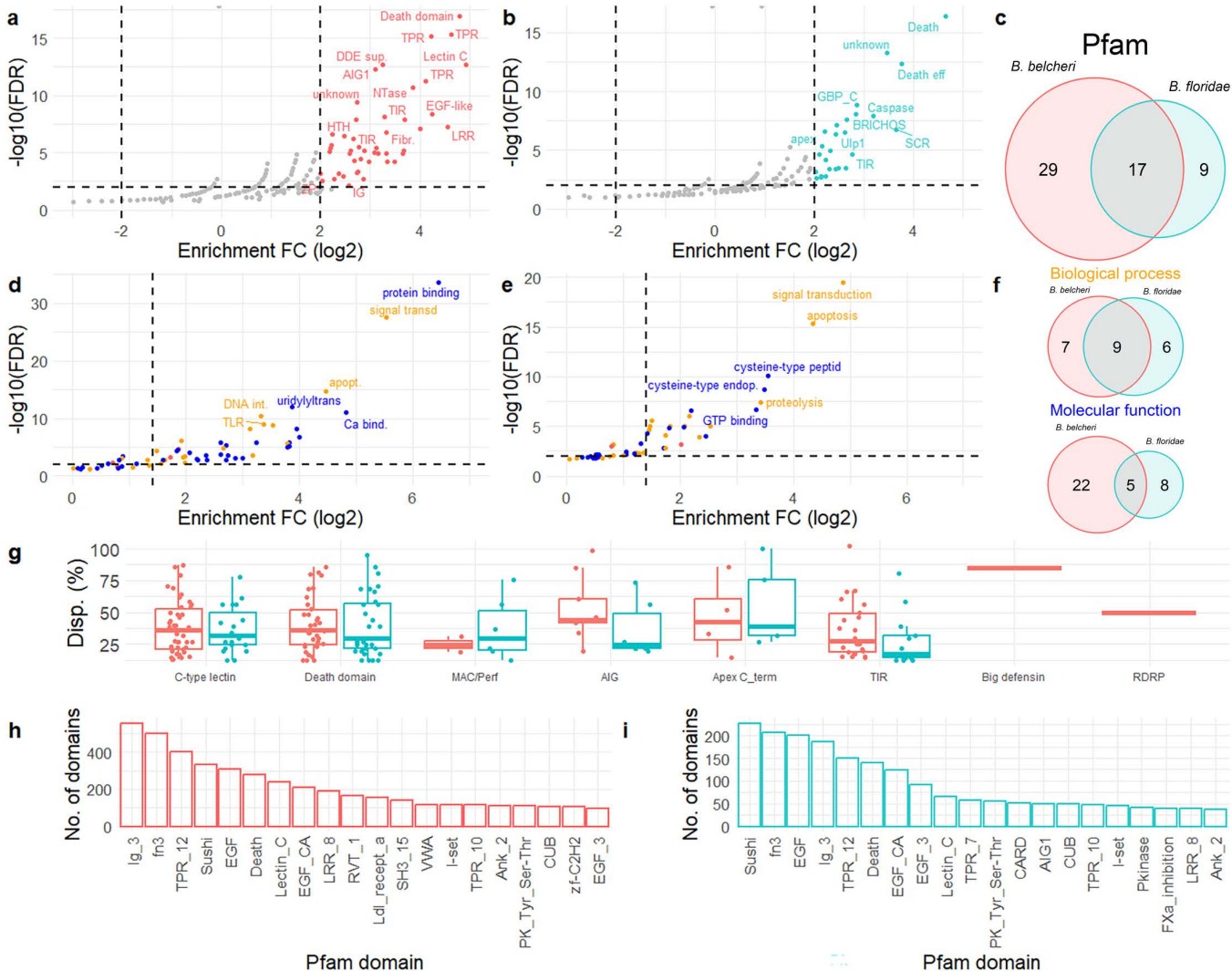

**Fig 4. Functional enrichment of dispensable genes in lancelets.** Volcano plot depicting the enriched Pfam domains encoded by the shell/cloud dispensable genes in *B. belcheri* (a) and *B. floridae* (b). The domains distribution is based on the false discovery rate (FDR) reported in a logarithm scale (-log₁₀) on the y-axis and the enrichment fold change reported as log₂ on the x-axis. Significantly enriched domains are coloured and named with abbreviations. c. Venn diagram depicting the enriched Pfam domains found exclusively in either *B. belcheri* or *B. floridae* or shared by the two species. Volcano plot depicting the enriched GO terms retrieved from the shell/cloud dispensable genes in *B. belcheri* (d) and *B. floridae* (e). The GO term distribution is reported as reported above for Pfam domains in panel (a); GO terms are coloured according to their classification into 'biological process' (orange), 'molecular function' (blue) or 'cellular component' (red). f. Venn diagram depicting the enriched biological process and molecular function GO terms found exclusively in either *B. belcheri* or *B. floridae* or shared by the two species. g. Dispensability fractions of selected gene families, reported as the number of individuals where the gene is absent over the total number of tested individuals. The abundance of the different Pfam domains encoded by the dispensable genes associated with de novo assembled NRSs is plotted for *B. belcheri* (h) and for *B. floridae* (i). The 20 most abundant domains are reported.

belcheri and *B. floridae*, respectively, even though some orthogroups were exclusively composed of dispensable genes (Fig 5c). Notably, the dispensable genes annotated in NRSs were distributed in 6,183 orthogroups, of which 55% were lancelet-specific.

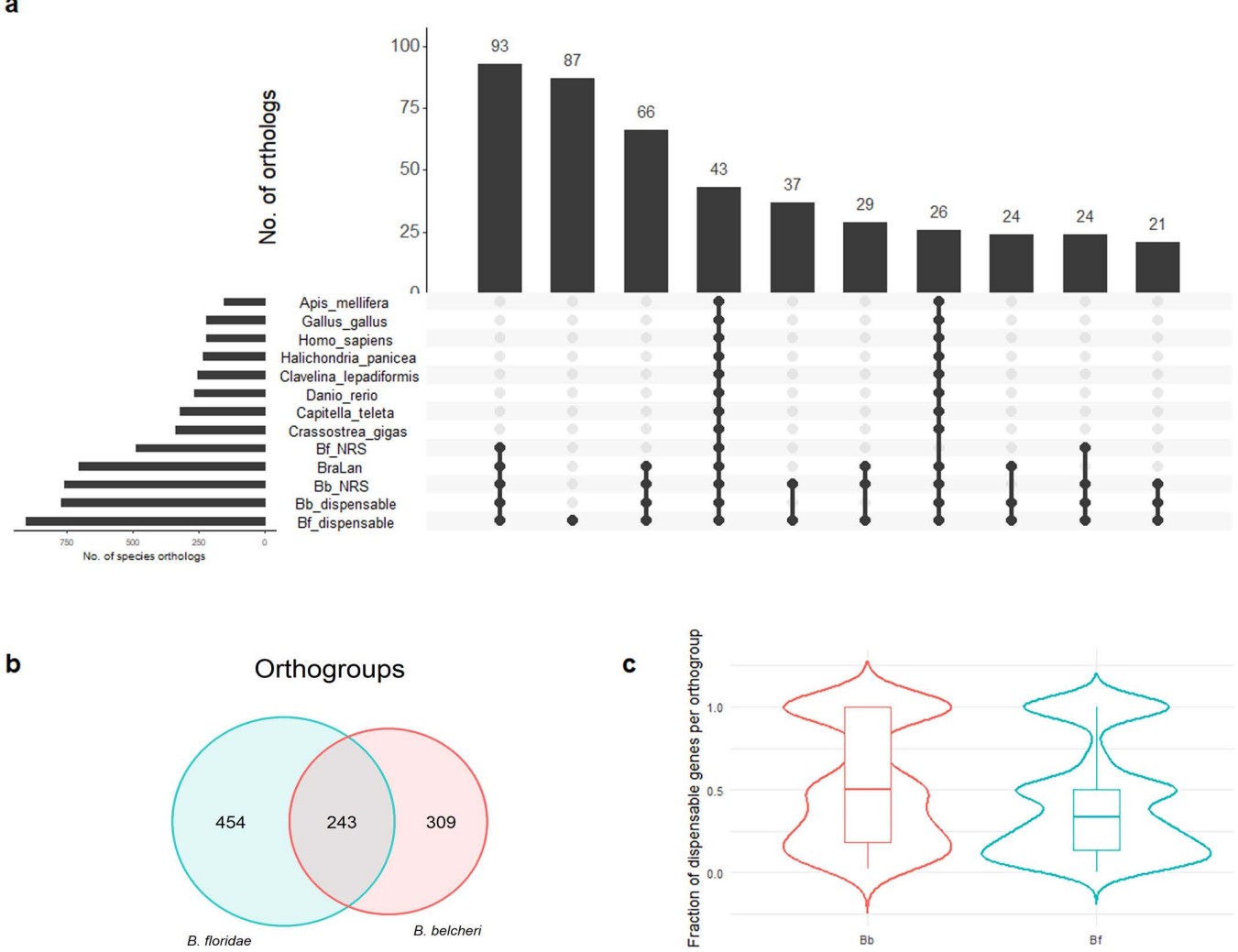

**Fig 5. Orthology analysis of lancelet dispensable genes. a.** Upset plot showing the distribution of the orthogroups that included lancelet dispensable genes and de-novo gene (NRS) among the 11 selected species. The graph reports the number of orthogroups detected for each species combination. **b.** Number of orthogroups exclusively found in *B. floridae*, *B. belcheri*, or in both species, among those including lancelet dispensable genes. **c.** Distribution of the fraction of dispensable genes per orthogroup in the two lancelet species.

## Dispensable defence genes in lancelets: GIMAP and APEC proteins as case studies

Our functional enrichment analysis revealed that the AIG1 domain was one of the most significantly over-represented Pfam domains among the dispensable genes in both lancelet species. This domain characterizes a large and heterogenous group of P-loop GTPases, which include animal GTPases of immunity-associated protein (GIMAP) family and plant immune-associated nucleotide-binding proteins (IAN) [36]. Indeed, 9 out of 13 GIMAP genes annotated in *B. belcheri* and 6 out of 12 GIMAP genes annotated in *B. floridae* were shell/cloud dispensable genes. We investigated in more detail the evolution of this gene family in lancelet, by recovering additional GIMAP genes from lancelet de novo assembled NRSs and comparing their sequences with those of the other species included in the orthology analysis reported in the previous section. Consistent with previous reports [37–39], GIMAPs have been subjected to multiple independent gene family expansion events during evolution, leading to the emergence of several lineage-specific clades (Fig 6a). This was also the case with

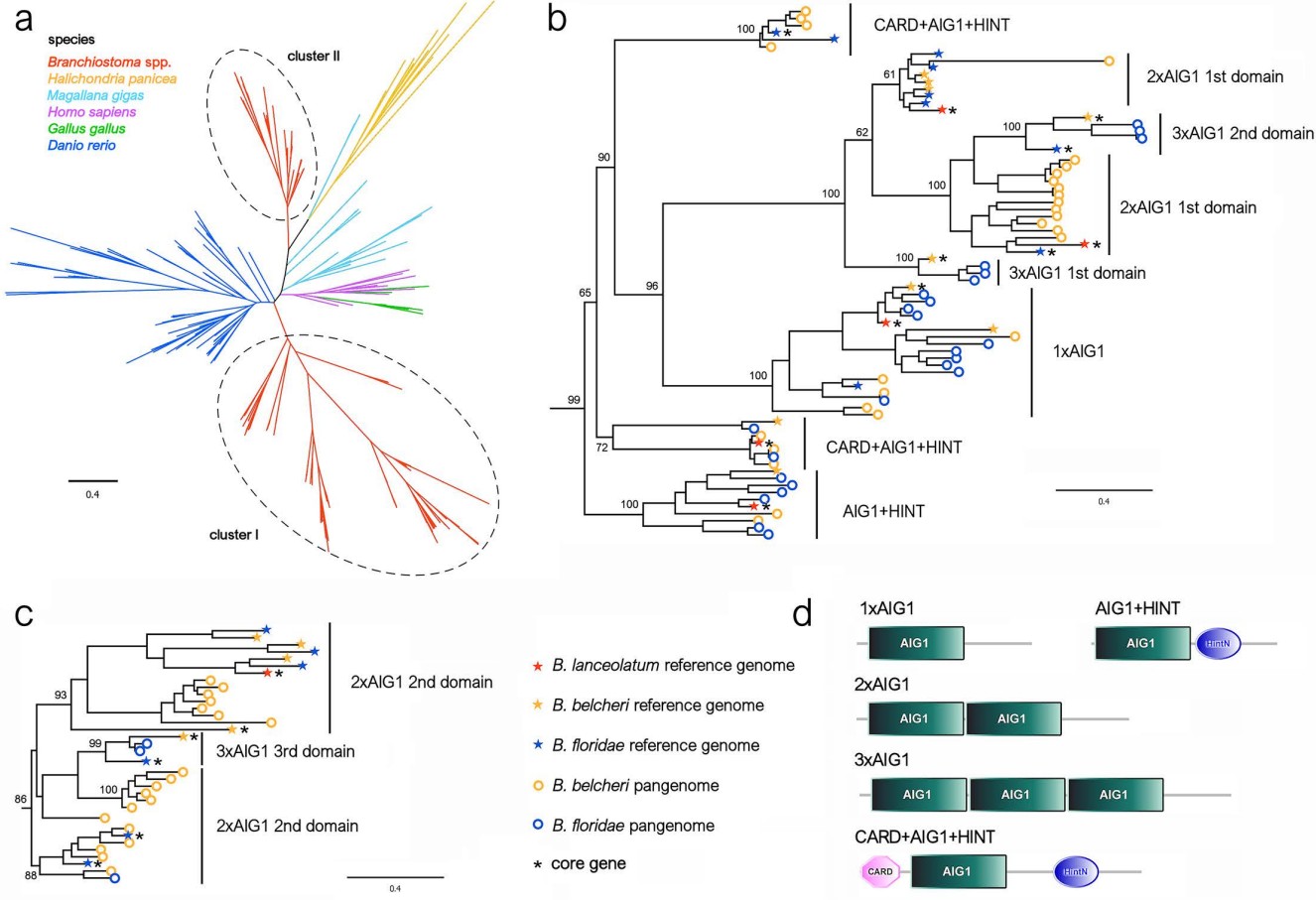

**Fig 6. The AIG1 domain.** a. Unrooted ML phylogeny of representative metazoan AIG1 family proteins. The lancelet proteins are highlighted in red and classified within two major groups, *i.e.,* clade I and clade II, which are detailed in panels b and c, respectively. Here, the bootstrap support values for the major nodes of the tree are shown, along with the core or dispensable status of the sequences and an indication of the domain architecture. d. Exemplified domain architecture of lancelet AIG1-domain containing proteins.

the lancelet sequences. In a phylogenetic tree based on the individual AIG1 domains present in multidomain proteins, the sequences formed two distinct clusters, distantly related to homologs from vertebrates, *M. gigas* and *H. panicea*. In particular, the two clusters represent: (i) a large group of AIG1 domains associated with different architectures (cluster I, Fig 6b); (ii) the C-terminal domain of proteins displaying either two or three consecutive AIG1 domains (cluster II, Fig 6c). The most recurrent architectures, which involved the presence of the N-terminal caspase activation and recruitment domain (CARD) and C-terminal HINT domains (Fig 6c), were unique to amphioxus among animals. However, some architectures, such as the CARD+AIG1+HINT domain combination, marked highly interesting cases of convergent evolution, being only present in a few multicellular fungi. Albeit unusual, similar occurrences have been previously documented for GIMAP-like genes in nematodes, where they have been interpreted as the probable result of horizontal gene transfer [38]. The coupling of the CARD and AIG1 domains further emphasizes the involvement of these proteins in apoptosis and inflammation.

The apextrin C-terminal (APEC) domain associated with pattern recognition receptors was also one of the most significantly enriched domains in the set of dispensable genes from lancelet, with 5 out of 13 annotated genes being subjected to PAV in both *B. belcheri* and *B. floridae*. We investigated the evolutionary history of these genes by evaluating their relationships with APEC-domain containing proteins from other metazoan lineages. In line with the results of a recent

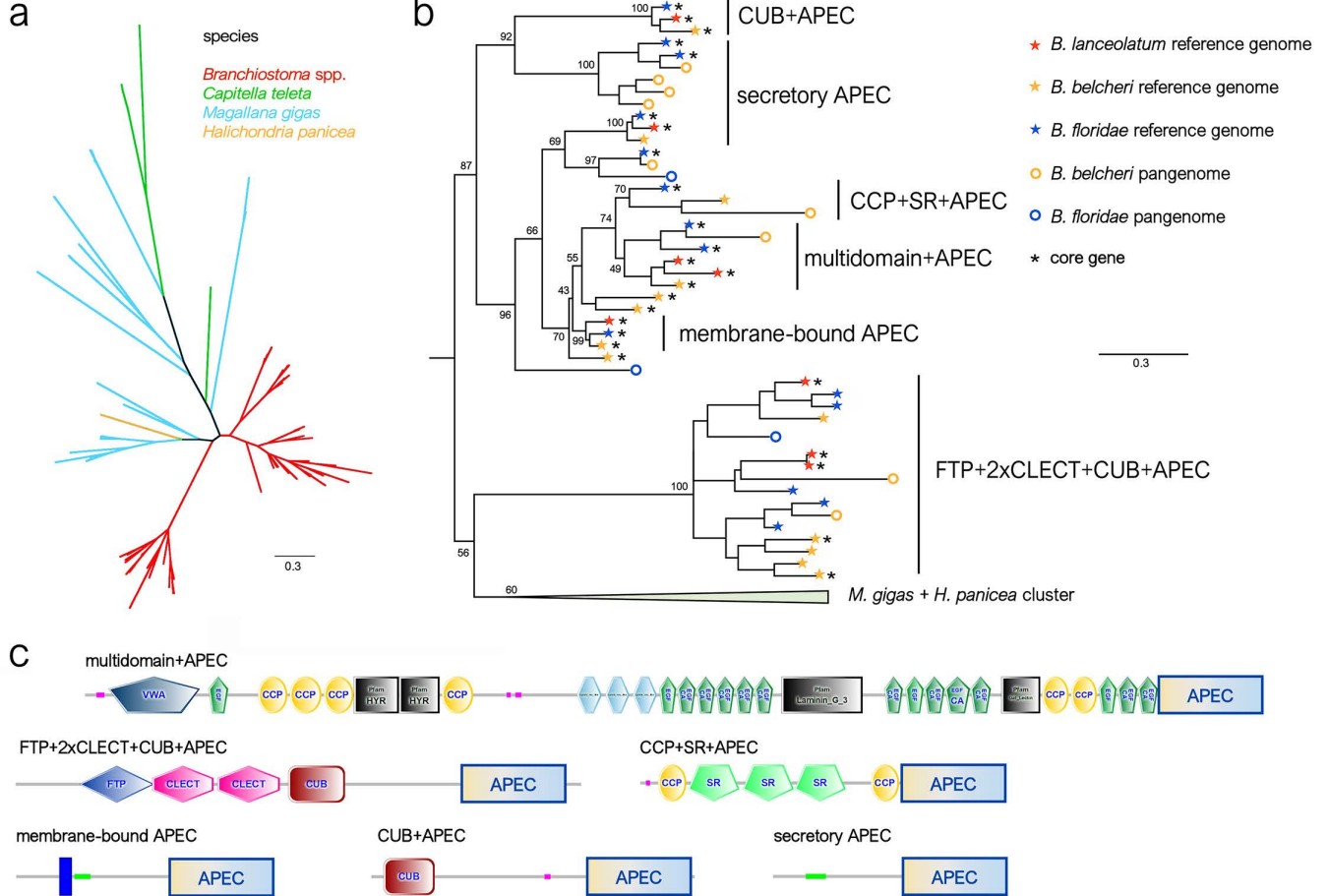

**Fig 7. The APEC domain. a.** Unrooted ML phylogeny of representative metazoan APEC family proteins. The lancelet proteins are highlighted in red. **b.** Detailed view of the phylogenetic relationship among lancelet APEC proteins. The bootstrap support values for the major nodes of the tree are shown, along with the core or dispensable status of the sequences and an indication of the domain architecture. **c.** Exemplified domain architecture of lancelet APEC-domain containing proteins.

evolutionary study, we could not detect any APEC domain containing proteins in either vertebrates or arthropods, and the sequences identified in the species selected for our comparative analyses were mostly grouped in lineage-specific clades, consistent with the previous report [40]. The lancelet APEC domain-containing proteins were grouped in two well-supported clades (Fig 7). The first clade included proteins carrying multiple carbohydrate recognition domains, namely, a fucolectin and two consecutive C-type lectin domains at the N-terminus, followed by a CUB domain (which, despite not being directly involved in pathogen recognition, is frequently associated with complement factors), and an APEC domain (Fig 7c). Strikingly, most of the sequences belonging to this clade were dispensable. The second clade was highly heterogeneous and included both simple secretory proteins [41,42] and more complex multidomain proteins. The identification of complex multidomain APEC domain-containing proteins was consistent with previous reports from mollusks [43], even though the domain combinations identified in amphioxus were different from those of mollusks (Fig 7b).

Beside these statistically enriched domains, we have also considered certain genes of interest. We note that one of the six cellular RNA-directed RNA polymerase (RdRP) genes reported in *B. belcheri* was also identified as dispensable in 50% of the tested individuals. In the same species, the big defensin gene was absent in 77% of the individuals, whereas the two arthropod-like defensin genes were not subjected to PAV (Fig 4g).

## Transgenerational dispensable gene transmission follows Mendelian inheritance rules

To gain insight into possible mechanisms of dispensable genes' transmission from parental genotypes to the offspring genomes, we analysed a dataset of 98 *B. floridae* samples, which included high coverage resequencing of two parents (namely M and F genotypes, ~60×) and low coverage resequencing of 96 offspring generated from these parents (~7×).

First, we investigated the M and F genotypes for the presence/absence status of the shell/core dispensable genes previously identified in *B. floridae* (Table 1). Using coverage cut-offs, we could establish the PAV status for 798 out of the 1,281 shell/core dispensable genes, whereas the remaining genes could not be reliably classified due to intermediate coverage values (S5 Fig).

A total of 282 genes annotated in the reference genome were absent in both parents (class I), whereas 311 genes found in hemizygosity in only one parental haplotype were missing in the other one (class II) and 137 genes were in hemizygosity in both F and M parents (class III). Moreover, 45 genes were present in only one parent (in association with both haplotypes) and 21 genes are present in one haplotype and in hemizygosity in the other one (Table 3). We calculated the fraction of genes missing in the offspring genomes for the three most abundant classes of genes, as well as for core genes. The finding that a median of 97% of the class I gene (i.e., genes absent in both parents, n = 282) were confirmed as missing in offspring samples confirmed the reliability of this approach (Fig 8). The other classes of genes were observed in offspring samples with frequencies close to those expected by Mendelian inheritance. In particular, those found in hemizygosity in only one of the two parental genotypes (class II) were absent in 40% of the offspring (expected 50%) and those found in hemizygosity in both parental genomes (class III) were absent in 19% of the offspring (expected 25%, Fig 8). The slight discrepancies between observed and expected PAV rates could be explained by the fact that read mapping was carried out against the reference genome assembly, which most certainly does not include a full collection of all the dispensable genes associated with *B. floridae*, thereby leading to the cross-mapping of reads deriving from similar dispensable paralogs.

## Discussion

It is becoming increasingly clear that multicellular eukaryotes, including animals, possess dynamic pangenomes, some of them with an architecture resembling that of the open pangenomes of bacteria [10,44]. Nevertheless, to date, animal pangenomes have been largely understudied, and the genomic mechanisms promoting large-scale structural variations,

**Table 3. Presence-Absence status of the *B. floridae* shell/cloud dispensable genes in F and M haplotypes and in the genomes of the progenies.** The analysis was carried out on 798 out of 1,281 shell/core dispensable genes previously identified in *B. floridae*. The gene class and its description in the parental genotypes, the expected offspring genotypes and absence percentages are reported together with the number of genes per category and the median of absence in offsprings.

| Gene class | Parental genotypes | | Expected offspring genotypes | | | Observed genes (No.) | Observed median absence rate in offspring |
|---|---|---|---|---|---|---|---|
| | F | M | ++ | +- | --(PAV) | | |
| core genes | ++ | ++ | 100% | / | / | 2 | / |
| I | -- | -- | / | / | 100% | 282 | 97.9% |
| II | +- | -- | / | 50% | 50% | 146 | 39.6% |
| | -- | +- | | | | 165 | |
| III | +- | +- | 25% | 50% | 25% | 137 | 17.7% |
| not tested | -- | ++ | / | 100% | / | 27 | / |
| | ++ | -- | | | | 18 | |
| not tested | +- | ++ | 50% | 50% | / | 7 | / |
| not tested | ++ | +- | | | | 14 | |
| undetermined | / | / | / | / | / | 420 | / |

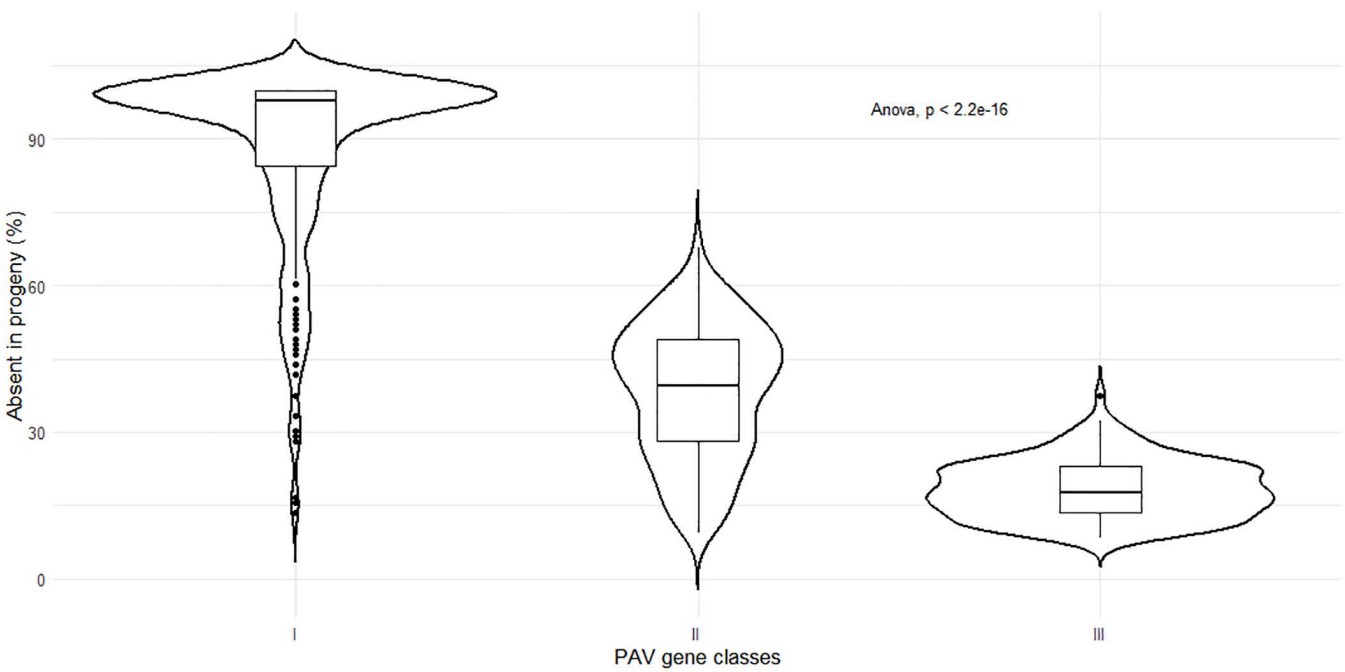

**Fig 8. Transmission of PAV genes from parents to offsprings.** The plot depicts the fractions of absent genes per sample (i.e., resequenced offspring) according to the four considered classes of *B. floridae* dispensable genes (see Table 2).

as well as the functional role played by the dispensable fraction of pangenomes are still debated [45,46]. Bivalves represent one of the best examples of animals with open bacterial-like pangenomes studied so far. In fact, following the first report of an open pangenome in the Mediterranean mussel [17], similar genomic organizations have been also reported in a number of other bivalve species [28,44]. Although still in their infancy, comparative pangenomic studies in metazoans indicate that the openness of vertebrate pangenomes is far lower than that of bivalves, which makes the investigation of pangenomes of early-branching chordate lineages particularly interesting in light of their evolutionary relatedness to vertebrates [31].

Surprisingly, our analysis of whole genome resequencing data revealed that hundreds of genes annotated in the *B. belcheri* and *B. floridae* reference genomes were dispensable, emphasizing the prevalence of structural variants subjected to PAV in lancelets. By analysing the genotypes of paired parent/offspring data in *B. belcheri*, we further demonstrated that the inheritance of lancelet dispensable genes conformed to Mendelian expectations, supporting the notion that, at least in this system, these genes behave as typical biallelic loci with presence/absence alleles. This result indicates that, despite theoretical possibilities for segregation distortion due to their association with structural variants, such effects may be minimal or absent in amphioxus. This would be consistent with the relatively small size of inferred indels associated with PAV in both lancelet species. The significant over-representation of Pfam domains and GO terms linked with intracellular pattern recognition, intracellular signalling, apoptotic and inflammatory processes in the dispensable fraction of lancelet pangenomes, is consistent with this interpretation. The strong representation of defence and immunity-related genes in lancelet pangenomes is further supported by a few key examples, such as the big defensin and eukaryotic RdRP gene families in *B. belcheri*. The single big defensin gene annotated in the reference genome, which encodes an antimicrobial peptide with broad-spectrum activity [47], was absent in 88% of *B. belcheri* samples, mirroring the findings previously reported for the same gene family in the Pacific oyster [48]. Although eukaryotic RdRPs (not to be confused with the non-homologous viral RdRPs), sporadically distributed in animals, are primarily known for their role in RNA interference,

this does not seem to be the case of lancelet orthologs [49], which may function in an as-yet undiscovered RNA-guided defence mechanism [50]. Strikingly, one of the six annotated RdRP genes of *B. belcheri* was absent in 50% of the resequenced individuals.

The fact that many of the dispensable genes are implicated in cellular defence and signalling is reminiscent of the "gene-accordion" of orthopoxviruses, a mechanism of rapid genome evolution through waves of gene duplication and diversification, followed by reduction in gene copy number, which allows these viruses to overcome host defences [51]. This process, apparently operating in most eukaryotic DNA viruses of the phylum *Nucleocytoviricota*, generates higher genomic entropy than simple SNPs, as reported in monkeypox virus [52,53]. Here we report that a similar "gene-accordion" operates in lancelets, with immunity-related genes undergoing dynamic gene copy number expansions, presumably in response to differential exposure to viral and/or bacterial pathogens. More broadly, the dynamic nature and high inter-individual variability of these genes appear to reflect a fundamental principle shared across all domains of life, from bacteria and archaea to eukaryotes, including cephalochordates, as demonstrated in this study. Indeed, prokaryotic immunity genes, including CRISPR-Cas systems, restriction-modification, Argonautes and many more, are usually part of the dispensable part of the pangenome, clustered in the hypervariable genomic loci [21,23]. Thus, in both prokaryotes and eukaryotes with open pangenomes, genomic innovations could be primarily stimulated by the interactions with diverse pathogens.

As instructive case studies, we focused our attention GIMAPs and APEC domain-containing proteins, both involved in defence mechanisms and subjected to extensive PAV in both lancelet species. Overall, we identified 21 novel GIMAP genes associated with dispensable NRSs in *B. floridae*, and 25 in *B. belcheri*. This would expand the complete repertoire of lancelet GIMAP genes to 38 and 33 in *B. belcheri* and *B. floridae*, respectively. Notably, GIMAPs have a complex evolutionary history, which has only very recently been investigated in a few metazoan phyla [37–39], with the overwhelming amount of functional information currently available only from human, where seven functional paralogous genes are present [54]. In humans, GIMAP genes appear to be mainly expressed in circulating immune cells, being crucial regulators of T-cell survival and differentiation [55,56], and their dysregulation is linked to several autoimmune diseases and cancer [57,58]. Although GIMAPs have been investigated in only a handful of other animal phyla, multiple lines of evidence support their involvement in immunity and stress tolerance. For example, GIMAPs have been hypothesized to play a key role in regulating hemocyte survival in oysters [59], which may be also supported by their strong association with PAV in mussels [17]. Another molluscan species, the snail *Biomphalaria glabrata*, has a largely expanded repertoire of GIMAPs, often expressed in the amebocyte-producing organ [60] and displaying marked differences in expression between individuals susceptible and resistant to *Schistosoma mansoni* infection [61]. The immune role of GIMAPs in animals has been proposed to be an ancestral feature, based on the evidence that these proteins retain their inducibility following PAMP stimulation in some cnidarian species [39,62].

APEC domain-containing proteins are a family of lectin-like secretory pattern recognition receptors (PRR), first described in *Branchiostoma japonicum* [41] and later also reported in *B. floridae*, where their involvement in microbial recognition, as well as in the modulation of immune response via the TRAF6/NF-kB axis, were confirmed [42]. Interestingly, recent studies have demonstrated that APEC domain-containing proteins with different structural organizations can act as PRRs in mollusks and echinoderms [43,63–65], and the presence of genes encoding similar proteins in the genomes of other metazoan phyla suggests that the APEC domain may have been recruited on multiple occasions as a carbohydrate recognition domain during evolution [66].

Phylogenetic evidence and gene PAV pattern analyses of GIMAPs and APEC domain-containing proteins suggest that these two gene families underwent rapid evolution in lancelets, leading to a high molecular diversification and contributing to the creation of novel protein forms most likely involved in immune recognition and signaling. We showed that most of the dispensable genes of lancelets belong to species- or family-specific clusters or orthologs, suggesting that PAV mostly impacted the recently acquired taxonomically-restricted gene families, or families that underwent lineage-specific

expansions, a phenomenon previously reported also for mussels [17]. Importantly, dispensable genes were, on average, expressed at lower levels than core genes, matching the observation collected for duplicated genes in *B. lanceolatum* [34], thereby suggesting that they can play a tissue-specific role, as expected for rapidly-evolving, young genes. Several orthogroups exclusively included dispensable genes, highlighting their functional significance for the establishment of individual gene complements and possibly also in the generation novel multi-domain proteins. Notably, a comparative genomic analysis of multi-domain proteins has revealed that *B. floridae* displays the highest number of novel domain combinations among eukaryotes [67], an observation which can now be at least in part explained as the widespread occurrence of PAV in this species.

Our pangenomic investigations also revealed the presence of a complete herpes-like virus genome (called BelcheriHV-1) integrated in the reference genome of *B. belcheri* and shared within 20% of the resequenced samples. A similar partial viral genome, called BelcheriHV-2, was found to be associated with a single *B. belcheri* sample. The integration of malacoherpes-like virus genomes has been previously reported in both lancelet species, but so far lacked validation [68–71]. Notably, mammalian herpesviruses (family *Orthoherpesviridae*) also can integrate into host genomes, as shown for the endogenized herpesvirus-6 found in ~1% of the human population, where it occupies different chromosomal positions and is associated with distinct polymorphisms [72]. Here, based on the comparison of proviral and host genome coverages, we suggest that, whenever present, BelcheriHV-1 may be found in either one or two copies per genome and seems to be poorly impacted by SNPs suggesting a recent integration. Furthermore, the identification of BelcheriHV-2 suggests that these viruses actively circulate in the lancelet populations. It has been previously suggested that *Alloherpesviridae* and *Malacoherpesviridae* shared a common ancestor [73,74], and thus the lancelet herpesviruses described herein could represent the missing link between invertebrate and vertebrate herpesviruses.

## Conclusions

Here, we reported the first pangenome analysis of lancelet species, revealing that the high heterozygosity typical of these species mirrors an extended open pangenome, with features resembling bacterial pangenomes. Although the high heterozygosity of these species and the lack of haplotype-resolved reference genome assemblies are limiting factors that could have, to some extent, decreased the sensitivity of our analyses and added biases related to inaccurate haplotype discrimination and/or overestimation of duplicated genomic regions [33,34], our investigation revealed the dynamic nature of the dispensable fraction of lancelet pangenomes. We demonstrated that dispensable protein-coding genes usually belong to highly variable multigene families involved in immune response and signalling processes. The proliferation of such genes appears to occur through duplication events, followed by the rapid molecular diversification of paralogs, yielding genomic regions with structural variation. Gene PAV can be seen as the readout of these processes or as a further mechanism to promote adaptation to changing environments and the exposure to pathogens. Further studies are needed to explore how specific combinations of dispensable genes may influence organismal fitness in lancelet species at both individual and population levels

## Materials and methods

### Data download and preliminary analyses

The reference genome assemblies of *B. floridae* (GCF_000003815.2) and *B. belcheri* (GCF_001625305.1) and their associated gene annotations were recovered from NBCI on December 1st 2023. Available whole genome resequencing data and related metadata of *B. floridae* (N = 41, plus N = 98 parental-offspring sequencing datasets) and *B. belcheri* (N = 100) were downloaded from the NCBI SRA database using the *srahunter* tool [75] (S1 Table). However, upon further inspection, one of the *B. belcheri* samples (i.e., SRR12010277) was identified as a misclassified *B. floridae* individual and thereby discarded, reducing the total number of suitable samples for this species to 99. A total of 91 and 61 RNA sequencing datasets of *B. belcheri* and *B. floridae* were also downloaded from the SRA archive (S1 Table). Raw reads were trimmed for quality and to remove adapter sequences using *fastp* v0.23.1 [76] with the following parameters: -V

--detect_adapter_for_pe -x -g -n 2 -5 -3 -p -l 75. The lancelet genomes were subject to a repeat prediction/classification and soft-masking step using *RepeatModeler v.2.0.5* and *RepeatMasker* v4.1.2 (http://www.repeatmasker.org/RepeatMasker/). The quality-trimmed DNA reads were mapped on the corresponding reference genome using *bwa mem* [77] allowing read multi-mapping and the resulting files were used for both SNP identification and for the subsequent PAV analysis. To identify SNPs, the mapping files were refined by removing duplicated reads, reads with more than 10 variants per 100-nt windows, and unaligned read ends (> 20 nt) using the tools implemented in the CLC Genomics Workbench v.23 suite (Qiagen, Hilden, Germany). SNP were called applying a minimal frequency of 35%, a minimum coverage of 10×, and subsequently annotated with overlapping information of genes and CDS to predict their functional consequences.

### Identification of dispensable regions and genes in *B. floridae* and *B. belcheri* genomes

To identify dispensable regions and genes in resequenced lancelet genomes we applied a previously developed PAV detection pipeline (https://github.com/Carmen-Tuc/PAV_pipeline) [78], with minor modifications. The average coverage of target regions (i.e., non-overlapping windows of genomic sequence spanning 5 kb) or genes (only considering exons), was calculated using samtools depth [79] by analyzing the bam mapping files obtained from each sample. Observed coverage values were compared to those expected for core regions, inferred from the analysis of single-copy complete Benchmarking Universal Single-Copy Orthologs (BUSCO) genes, identified using BUSCO v5.2.2 [80] with the metazoa_odb10 dataset. Absent regions or genes were defined as those with coverage below one-eighth of the expected value. An arbitrary threshold, set at 10% of the samples analysed for each of the two species, was applied to identify shell and cloud dispensable genes and regions to be subject to further analysis, thereby excluding softcore elements present in the majority (i.e., >= 90%) of individuals. The percentage of mapped reads per sample was determined with *samtools stats* and used to detect possible outliers. The presence/absence matrix of dispensable regions was used to run a Principal Component Analysis (PCA) using the R packages *FactoMineR*, *ggcorrplot*, *corrr* and *factoextra*, and to build a heatmap with the *heatmaply* package. *B. belcheri* and *B. floridae* proteomes were functionally annotated using *InterProScan* v.5.57-90.0 [81]. An enrichment analysis was computed with a hypergeometric test based on the Pfam domains and on the GO terms associated with shell and cloud genes using an in-house python script available at https://gitlab.com/54mu/enrichment_test [82]. Annotations were considered significantly enriched if they exhibited an observed-to-expected count ratio greater than 4 for Pfam domains or greater than 2 for GO terms, and if the associated False Discovery Rate (FDR)-corrected P-value [83] was lower than 0.01.

The genomic distribution of dispensable genes was further inspected in *B. floridae*, by calculating the density of dispensable genes per Mb and graphically representing the data with Circos [84]. To assess the correlation between the proximity between dispensable pairs and their PAV patterns, Wilcoxon rank-sum tests were used to evaluate statistically significant differences (i.e., FDR-corrected p-value < 0.05) between intra- and inter-chromosomal gene pairs (with the latter used as a background, due to the expected lack of correlation), computed as the average value for 5 Kb-long intervals of distance.

In addition, the possible association between dispensable regions and flanking repeated elements was investigated by comparing the repeat content in the upstream and downstream regions of each dispensable gene block (obtained from the libraries generated by RepeatModeler) to that of all core genes. The significant enrichment (i.e., FDR-corrected p-value < 0.05) for each category of repeats found in upstream and downstream flanking regions (spanning 5, 10 or 20 Kb of genomic sequence) was assessed with a hypergeometric test.

The *B. floridae* parental-offspring sequencing dataset was further analyzed to determine the heterozygosity or homozygosity status of shell/cloud dispensable genes in the female (F) and male (M) parental genotypes. Coverage-based thresholds were established using the average read depth of genes not affected by presence/absence variation (PAV). Genes in hemizygous state were defined as those with coverage within ±50% of half the average depth (i.e., 15–45× for the F genotype and 14–42× for the M genotype). All statistical analyses, data manipulation and visualisation steps were performed in R 4.2.3 [85] using the *tidyverse*, *ggplot2*, *ggpubr*, *ggrepel*, *VennDiagram*, *UpSetR*, *smplot2* and *data.table* packages. The script is available at https://github.com/umberos/Lancelet_pangenome.

## Analysis of the de-novo assembled pangenome and of associated non-reference sequences

For each resequenced lancelet individual, the DNA reads that could not be mapped on the reference genomes were recovered from the mapping files using *samtools* and de novo assembled with *SPAdes v3.15.5* [86], applying the *–isolate* flag. Assembled non-redundant contigs longer than 2.5 kb (obtained with *cd-hit-est* 0.8 [87]) were considered as NRSs and were therefore subject to gene prediction using two different approaches, namely *Prodigal v2.6.3*, run in metagenomic mode, and *BRAKER v3.0.8* [88], using the *B. floridae* gene models as training set. The non-redundant coding sequences predicted by *BRAKER3* were translated into proteins, subjected to another round of redundancy removal (*cd-hit* 0.8), and taxonomically classified using *diamond v. 2.0.6 blastp* searches against the NCBI *nr* database [89], only keeping those assigned to *Metazoa* for subsequent analyses. Viral hits among the assembled contigs were detected using *geNomad v1.7.4* [90], based on *prodigal* gene predictions and viral marker protein profiles. The hits showing relevant viral matches were extracted and further analysed. *Blastp* searches using the relevant translated ORFs as queries were performed against the NCBI nr database (downloaded February 2023) to assign a best match to each hit.

## Gene expression analysis

Quality-trimmed RNA reads were mapped on the corresponding reference genomes using the *CLC mapper* of the CLC Genomics Workbench, applying the following parameters: mismatch cost = 2; insertion cost = 3; deletion cost = 3; length fraction = 0.8; similarity fraction = 0.8 and expression values were counted as Transcripts Per Million (TPM). The computed expression levels, as well as the fraction of samples without detectable transcription, were compared between the sets of core and shell/cloud genes.

## Gene orthology analysis

The proteomes of 11 species including the vertebrates *Homo sapiens*, *Danio rerio* and *Gallus gallus*, the ascidian *Clavelina lepadiformis*, the invertebrates *Crassostrea gigas*, *Capitella teleta*, *Apis mellifera* and *Halichondria panicea*, three lancelet species (*B. belcheri*, *B. floridae* and *B. lanceolatum*) and the proteins encoded by the de novo assembled NRSs of lancelets were processed to identify ortholog genes using *orthofinder v2.5.4*, with *mafft* and *FastTree* used to infer maximum likelihood trees from multiple sequence alignments [91]. The orthogroups that included lancelet dispensable genes were identified and the number and origin of orthologs in these clusters were counted. To investigate specific domains, all protein sequences derived from the annotation of the three lancelet reference genomes that included at least one AIG1 (PF04548) or APEC (PF16977) domain were extracted, together with the homologous sequences placed in the same orthogroups from other selected metazoan species. Due to the highly variable architectures of the protein sequences, only the region comprising the characterizing domain was selected for phylogenetic inference analysis. In detail, all accessory portions of sequences present either at the N-terminal or at the C-terminal end of the target domains were removed, and only sequences covering at least 50% of the expected length were kept for subsequent analysis. Whenever multiple domains were detected, all of them were extracted and added to the list. Trimmed sequences were aligned with MUSCLE [92] and, upon manual curation, which removed all the positions of the alignment with gaps in > 50% entries, the multiple sequence alignments were subjected to maximum likelihood phylogenetic analyses with Iq-Tree [93]. The best-fitting molecular models of evolution, according to the corrected Bayesian information criterion, were WAG + F + R5 and WAG + F + R3 for the AIG1 and APEC datasets, respectively. The reliability of the trees was evaluated using an ultrafast bootstrap test with 1000 replicates.

## Analysis if the integrated BelcheriHV-1 genome

To confirm the presence of the BelcheriHV-1 genome in the 20 samples showing the presence of the 35 contiguous viral genes, *de novo* assemblies were performed for each sample, and the resulting contigs were subsequently mapped to the *B. belcheri* reference genome using the *large gap mapper* tool implemented in CLC Genomics Workbench. The mapping files were manually inspected to determine the coverage of the BelcheriHV-1 genome.

PLOS Genetics

# Supporting information

**S1 Table. Metadata of the datasets analysed for this study.** The list included the NCBI IDs and relevant metadata for the genomic and transcriptomic datasets analysed in this study.
(XLSX)

**S2 Table. Size distribution of the blocks of contiguous shell/cloud dispensable genes in *B. belcheri* and *B. floridae*.** *This gene block corresponds to the integrated BelcheriHV-1 genome.
(XLSX)

**S3 Table. Significantly enriched (hypergeometric test, FDR-corrected p-value) repeated elements associated with shell/core gene blocks, compared with core genes.** Enrichments tests were carried out for flanking regions of different sizes (i.e., 5, 10 and 20 Kb).
(XLSX)

**S4 Table. Annotation file of the BelcheriHV-1 and BelcheriHV-2 predicted proteomes.** The table reported the *blastp* best hit against the NCBI non-redundant protein database (nr) and against the SWISS-prot database.
(XLSX)

**S5 Table. Enrichment test performed for *B. belcheri* and *B. floridae* shell/cloud dispensable genes.** For both Pfam domains and GO terms, the ID and description, the total counts in the predicted proteome, the observed counts among dispensable genes, the FDR p-value and the enrichment value are reported. For GO terms, the classification into biological and molecular functions are also reported.
(XLSX)

**S1 Fig. Correlation between gene dispensability and the number of observed PAV occurrences among the resequenced samples of *B. belcheri* (a) and *B. floridae* (b).**
(TIFF)

**S2 Fig. Average concordance (± SD) between the presence/absence status of intrachromosomal pairs of dispensable genes in *B. belcheri*, for bins of pairwise distance ranging 5Kb each.** Statistically significant differences (Wilcoxon rank-sum tests, FDR-corrected p-value < 0.05) compared with inter-chromosomal dispensable gene pairs (dashed red line) are highlighted by an asterisk. Note that the genes included in the integrated BelcheriHV-1 were omitted.
(TIF)

**S3 Fig. Average concordance (± SD) between the presence/absence status of intrachromosomal pairs of dispensable genes in *B. floridae*, for bins of pairwise distance ranging 5Kb each.** Statistically significant differences (Wilcoxon rank-sum tests, FDR-corrected p-value < 0.05) compared with inter-chromosomal dispensable gene pairs (dashed red line) are highlighted by an asterisk.
(TIF)

**S4 Fig. Principal Component Analysis and sample correlation plots based on the presence-absence patterns of shell/cloud dispensable genes for *B. belcheri* (a-b) and *B. floridae* (c-d).** The *B. belcheri* samples in the PCA are coloured according to the presence of the integrated BelcheriHV-1 genome (YES: blue, NO: red). Instead, the B. *floridae* samples in the PCA are coloured by experiment, with the blue ones referring to the parental + offspring dataset.
(TIF)

**S5 Fig. Genome coverage plot of the female (a) and male (c) genotypes used to produce the offsprings.** The orange dotted lines indicated the average coverage computed based on core genes. Gene coverage plots for the

dispensable genes in the female (b) and male (d) genotypes. The pink dotted lines indicate the cut-off used to determine the absence of a given gene, whereas the two blue lines indicate the range used to determine the genes in hemizygosity, and the orange lines refer to the expected coverages of core genes.
(TIFF)

**S1 File.  The data matrix reporting the presence-absence of *B. belcheri* genes in the tested DNA samples.** "1" refers to an absent gene, "0" to a present gene.
(CSV)

**S2 File.  The data matrix reporting the presence-absence of *B. floridae* genes in the tested DNA samples.** "1" refers to an absent gene, "0" to a present gene. The matrix included also the dataset of the offspring, for a total of 140 samples.
(CSV)

**S3 File.  BelcheriHV-1 genome (fasta format).**
(FA)

**S4 File.  BelcheriHV-2 genome (fasta format).**
(FA)

## Acknowledgments

Computational resources were provided by the University of Padova Strategic Research Infrastructure Grant 2017: *CAPRI: Calcolo ad Alte Prestazioni per la Ricerca e l'Innovazione*.

## Author contributions

**Conceptualization:** Umberto Rosani, Marco Gerdol, Mart Krupovic.

**Data curation:** Umberto Rosani.

**Formal analysis:** Umberto Rosani.

**Funding acquisition:** Umberto Rosani, Marco Gerdol.

**Methodology:** Umberto Rosani, Marco Gerdol.

**Supervision:** Marco Gerdol, Mart Krupovic.

**Writing – original draft:** Umberto Rosani.

**Writing – review & editing:** Umberto Rosani, Marco Gerdol, Mart Krupovic.

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
