## [Decision Letter · Decision Letter 0]

4 Mar 2025

PGENETICS-D-25-00112

The highly dynamic pangenome of basal chordates is enriched in defence and immunity genes and is inherited following the Mendelian law

PLOS Genetics

Dear Dr. Rosani,

Thank you for submitting your manuscript to PLOS Genetics. After careful consideration, we feel that it has merit but does not fully meet PLOS Genetics's publication criteria as it currently stands. In particular, both reviewers commented that the logical flow of the manuscript is difficult to discern and it is written for the most expert reviewer rather than a general readership. There are also some important technical concerns; please also see the comment about code accessibility. In light of these comments, we invite you to submit a revised version of the manuscript that addresses the points raised during the review process.

Please submit your revised manuscript within 60 days May 03 2025 11:59PM. If you will need more time than this to complete your revisions, please reply to this message or contact the journal office at plosgenetics@plos.org. Please include the following items when submitting your revised manuscript:

We look forward to receiving your revised manuscript.

Kind regards,

Harmit S. Malik

Academic Editor

PLOS Genetics

Anne Goriely

Editor-in-Chief

PLOS Genetics

Aimée Dudley

Editor-in-Chief

PLOS Genetics

Anne Goriely

Editor-in-Chief

PLOS Genetics

**Additional Editor Comments (if provided):**

**Journal Requirements:**

**Reviewers' comments:**

Reviewer's Responses to Questions

**Comments to the Authors:**

Reviewer #1: This manuscript presents a study of the gene repertoire in the amphioxus species B. belcheri and B. floridae. They use publicly available genomic and transcriptomic data from a large number of B. belcheri and B. floridae individuals. They study presence-absence variation (PAV) and non-reference sequences (NRS) in a mainly gene-centric approach to gain insight into the genes that are dispensable in these species.

Amphioxus species are known to have an extremely high level of heterozygosity, and thus studying genomic diversity in an alternative way by studying the genes that are dispensable may shed light on this extreme case of genomic diversity at the base of the chordate tree. However, the present manuscript does not fulfil the expectations expressed in the title and abstract, nor does it meet proper scientific publishing standards in its current form. A detailed revision of the manuscript in its current form cannot be done properly. Several important issues would need to be addressed beforehand.

First, the text is long and complicated. It is difficult to follow and lacks rationality and structure in many places. The whole manuscript needs a thorough rewrite with the aim of making it more concise and understandable to an expert in the field. Please take the time to improve the text, give each paragraph a message and link them to an overall rationale.

Second, the results are a compendium of figures and statistics without a clear rationale or link between them. In other words, they lack logic and structure. In addition, the text does not really address all the results shown in the figures, and it is not clear which species or test is being referred to at each point. Figure labels and captions do not always stand alone. Please ensure that the main figures show the reader what they need to see to understand the results and the rationale of the manuscript. Make sure that the figures and their captions are self-explanatory. Consider adding a schematic figure explaining the methods and classification of regions/genes as core, shell/cloud or dispensable.

Third, the term "pangenome" is not used in the same way in all studies (see Matthews CA et al., A gentle introduction to pangenomics. Brief Bioinform. 2024); its use has evolved since it was first used in the early 2000s with the advent of new sequencing technologies and the availability of data. In this study, the authors use a classic gene-centered approach rather than a fully de-novo assembly approach, although they do re-assemble the unmapped genes to include non-reference genes. I agree that this is a valid and informative approach given the available data, but I would like to see all this clarified and discussed in the manuscript.

Fourth, at several points in the text, the authors seem to argue that it is not universal for all species to have a pangenome, but that some species might not have one (see, for example, the first subtitle of results “PAV analysis reveals the existence of a pangenome in two lancelet species”). Please develop how a given species might not have a pangenome.

Finally, as a bioinformatic work, I would like to see all the code used to get the results in a public repository, such as GitHub.

Reviewer #2: This paper by Rosani et al. describes the assembly and analysis of the pangenomes of two closely related cephalochordates, Branchiostoma floridae and belcheri. The starting point are the publicly available genome assemblies from both species and genomic read libraries from a large number of individuals of each species. Using their own presence-absence variation (PAV) analysis pipeline the author then identified hundreds (!) of genes that are either present or absent in individuals of each population for both species. This is then followed by classifying them based on common features of the encoded protein including gene ontology and domain content. Intriguingly, but may be not surprisingly, the main commonality that emerges (consistently in both species) is that they appear to encode genes with predicted immune function. This is in line with the widely held view that variations of immune genes within a population are important features to allow for the survival of species in light of the rapidly evolving microbial universe from which new pathogens can easily arise. The author here now demonstrate clearly that it is not just SNPs but also the PAVs that contribute to this intraspecies genetic diversity. This manuscript is of high interest for a very broad range of scientists including but not limited to those who study structural features of genomes and comparative immunologists. It is a very important piece of work that will gather significant attention.

1 ) My main concern is that the beginning of results section uses too much bioinformatics jargon and is too densely written such that it is hard and tedious for the broad audience to keep the attention until they later on can identify with domain and gene names that are familiar. I recommend recruiting the help of a wet lab molecular biology colleague to revise the text such that it becomes more easily accessible – it will be worth it.

2 ) After reading this manuscript many immunologist will likely wonder about whether the Branchiostoma RAG1/2-like genes (thought to a the ancestor of the V(D)J recombinase) are among the genes with PAV or not. A short note on this would be good to include.

3 ) What is not investigated in depth are details of the chromosomal locations where PAV occur. It would be interesting to see whether the sequences surrounding gene gain and loss show common features (e.g. transposon mediated integration would lead to target site duplication). Even if a detailed analysis would likely be beyond the scope of this manuscript, the authors could may be add a few sentences in the discussion about the mechanisms by which they envision these events to occur.

**Have all data underlying the figures and results presented in the manuscript been provided?**

Reviewer #1: Yes

Reviewer #2: Yes

PLOS authors have the option to publish the peer review history of their article (what does this mean? ). If published, this will include your full peer review and any attached files.

**Do you want your identity to be public for this peer review?** For information about this choice, including consent withdrawal, please see our Privacy Policy .

Reviewer #1: **Yes: ** Marina Brasó-Vives

Reviewer #2: **Yes: ** Sebastian D Fugmann

**Figure resubmission:**
---

## [Decision Letter · Decision Letter 1]

15 Jul 2025

PGENETICS-D-25-00112R1

The highly dynamic pangenome of basal chordates is enriched in defence and immunity genes and is inherited following the Mendelian law

PLOS Genetics

Dear Dr. Rosani,

Thank you for submitting your manuscript to PLOS Genetics. After careful consideration, we feel that it has merit but does not fully meet PLOS Genetics's publication criteria as it currently stands. However a careful revision along the lines suggested by Reviewer 1 would satisfy this criteria. Therefore, we invite you to submit a revised version of the manuscript that addresses the points raised during the review process.

Please submit your revised manuscript within 30 days Aug 14 2025 11:59PM. If you will need more time than this to complete your revisions, please reply to this message or contact the journal office at plosgenetics@plos.org. Please include the following items when submitting your revised manuscript:

We look forward to receiving your revised manuscript.

Kind regards,

Harmit S. Malik

Academic Editor

PLOS Genetics

Anne Goriely

Editor-in-Chief

PLOS Genetics

Aimée Dudley

Editor-in-Chief

PLOS Genetics

Anne Goriely

Editor-in-Chief

PLOS Genetics

**Additional Editor Comments (if provided):**

**Journal Requirements:**

**Reviewers' comments:**

Reviewer's Responses to Questions

**Comments to the Authors:**

Reviewer #1: The overall manuscript has improved greatly after the revision. The manuscript is now much more readable and the results and conclusions stated more logically. I appreciate the effort by the authors. As stated previously I find this work interesting, adequate and an important contribution to amphioxus and metazoan genomics. Having said this, I find several major and minor issues still need to be addressed.

Major comments:

- I find the figures are still not always properly framed and interpreted in the results section. The results describe the overall outcome of each figure. However, they sometimes fail to fully explain the details that can be seen in the figure, and they frequently leave the reader to interpret and guess the consequences. I left comments on the most important cases as minor comments, but I would appreciate an overall effort to revert this tendency.

- Is it expected that the inheritance of PAV would follow Mendelian patterns of inheritance? What would be the alternatives? Please develop on this in the introduction and discussion sections.

- The use of the term “pangenome” in some sections of the manuscript is still confusing. It is used as synonym to “dispensable part of the genome" and in some sentences it can still be interpreted that not all species have by definition a pangenome. This is misleading and in some cases, it can lead to misunderstandings. I point the most critical cases as minor comments, but I would appreciate it if the authors would carefully read all the text once more, being attentive to this.

- I understand that the methods are very explicit and that the PAV analysis and GO/Pfam enrichment use already published pipelines with few modifications. However, I must insist on publication of the full code used in the study in order to make the research in this manuscript fully reproducible. The use of common R packages should not excuse authors from publishing the code that uses them. Commands executed on the data previously or between pipelines can affect the results significantly, and the code to generate the figures is very important for their reproducibility. The manuscript would benefit greatly if the code to reproduce the analysis and figures is made available via a public repository. This is currently the standard of good practice in bioinformatics.

- Finally, a thorough re-reading with a critical eye would greatly improve the text's cohesion and clarity.

Minor comments:

L106: B. lanceolatum also has a chromosome level assembly (see reference 65)

L109: Please delete “presence of” since what this study investigates is not the presence of a pangenome but its architecture.

L145: Please cite Table S1 for the reference of the transcriptomic datasets.

T1: Change “No. of datasets” to “No. of samples” or “No. of individuals” which are more clear.

F1b: The scale of the axes does not allow for the appreciation of the correlation. Please zoom in. Also, for B. belcheri there are two samples (presumably the two samples used to generate the reference genome assembly) that clearly cluster separately from the others and drive the correlation. Please, specify which samples they are to justify why they cluster apart. If they are the two reference samples, consider calculating the strength of the B. belcheri correlation without them, since they are a clear exception to the analysis.

L164: B. floridae is not the only species with chromosome level assembly. See L106 and reference 65.

F3b: Please rephrase the caption and associated text in results and add a more clear explanation. It is not obvious to understand what the figure shows and means. Also, consider changing figure labels to indicate that they refer to “coverage” ratios.

F3c & d: In the results text, they are wrongly referenced (F3c must be F3d and vice versa).

F3c & d and related text: It is not clear what “SNP frequency” refers to. See, for example, the caption of the figure where “SNP frequency” is used to describe both the prevalence of given SNPs among samples (F3c y axis) and the distance between SNPs along the genome/virus (SNP every x nt in F3b y axis). Please clarify in the results text, figure caption and figure labels.

F3c: I don’t understand what F3c is representing and what its message is. The results text says that all viral SNPs appear mostly at a frequency of 1 or 0.5 (across samples). How can SNP frequency across samples be plotted per sample? Please rephrase and clarify both in the results text and figure caption. Also, a boxplot is not appropriate to show such extreme distributions of frequencies. Please, find another way of representing it. Why do two samples look so different from the rest? Please, explain in the results text how to interpret the figure results.

F3d: What do the lines linking the points on one boxplot and the other represent?

T2 & T3: T3 is referenced before T2. Please, change labels accordingly.

L246: Please, develop a bit further.

L261: Please change “pangenomes” to “dispensable genes”. What this section analyses are the potential functional consequences of the dispensable genes, not the entire pangenome which includes the whole genome.

L265: Please develop further on the results. What type of Pfam domains are enriched in dispensable genes?

L267: Please change the sentence starting at this line to something like: “a and b. Volcano plot depicting the enriched Pfam domains encoded by the shell/cloud dispensable genes in B. belcheri (a) and (b) B. floridae”. Same for the sentence starting at L272 referencing figures F4d & F4e and at L 279 for figures F4h & F4i (note the formatting in this last one).

L320: I don’t see how this is the conclusion of such a paragraph. Please rephrase or if it is intended to be a conclusion of the entire section, please make it clear.

F6b & F6c: Why are only reference sequences labeled as dispensable? Aren’t the further sequences discovered while building the pangenome also dispensable by definition? Please clarify in the figure or text or explain if I’m getting something wrong. Same for F7b.

F6d: This figure needs a better contextualization and labeling to link it to the labels in figures b and c. Same for

F7c.

F8a-d: These are very methodological figures. They show no results. Please, move them to the supplementary materials.

L422: Please, cite F8e in the sentence that finishes here.

L435: Please rephrase. This sentence implies that not all species necessarily have a pangenome.

L544: Please discuss why you think that gene duplication and exon shuffling are the main cause of the generation of the dispensible part of the genome. Are your results suggesting something like this? Isn’t virus genome integration another mechanism that your results suggest having had a role in it?

Reviewer #2: The manuscript has clearly improved from the previous version in terms of readability, and it will hopefully attract the broad readership of PLoS Genetics to a research area that has been largely ignored thus far by focussing on lab model organisms that exhibit rather limited genetic diveristy (if at all).

**Have all data underlying the figures and results presented in the manuscript been provided?**

Reviewer #1: Yes

Reviewer #2: Yes

PLOS authors have the option to publish the peer review history of their article (what does this mean? ). If published, this will include your full peer review and any attached files.

**Do you want your identity to be public for this peer review?** For information about this choice, including consent withdrawal, please see our Privacy Policy .

Reviewer #1: **Yes: ** Marina Brasó-Vives

Reviewer #2: **Yes: ** Sebastian D Fugmann

**Figure resubmission:**
---

## [Editor Report · Decision Letter 2]

5 Aug 2025

Dear Dr Rosani,

We are pleased to inform you that your manuscript entitled "The highly dynamic pangenome of basal chordates is enriched in defence and immunity genes and is inherited following the Mendelian law" has been editorially accepted for publication in PLOS Genetics. Congratulations!

Yours sincerely,

Harmit S. Malik

Academic Editor

PLOS Genetics

Anne Goriely

Editor-in-Chief

PLOS Genetics

Aimée Dudley

Editor-in-Chief

PLOS Genetics

Anne Goriely

Editor-in-Chief

PLOS Genetics

Comments from the reviewers (if applicable):

**Data Deposition**

http://datadryad.org/submit?journalID=pgenetics&manu=PGENETICS-D-25-00112R2

**Press Queries**

---

## [Editor Report · Acceptance letter]

PGENETICS-D-25-00112R2

The highly dynamic pangenome of basal chordates is enriched in defence and immunity genes and is inherited following the Mendelian law

Dear Dr Rosani,

We are pleased to inform you that your manuscript entitled "The highly dynamic pangenome of basal chordates is enriched in defence and immunity genes and is inherited following the Mendelian law" has been formally accepted for publication in PLOS Genetics! Your manuscript is now with our production department and you will be notified of the publication date in due course.

With kind regards,

Zsofia Freund

PLOS Genetics

On behalf of:
